# Molecular basis of wax-based color change and UV reflection in dragonflies

Ryo Futahashi[1]*, Yumi Yamahama[2], Migaku Kawaguchi[3], Naoki Mori[4], Daisuke Ishii[5], Genta Okude[1,6], Yuji Hirai[7], Ryouka Kawahara-Miki[8], Kazutoshi Yoshitake[9], Shunsuke Yajima[8,10], Takahiko Hariyama[2], Takema Fukatsu[6,11]

[1]Bioproduction Research Institute, National Institute of Advanced Industrial Science and Technology (AIST), Tsukuba, Japan; [2]Department of Biology, Hamamatsu University School of Medicine, Shizuoka, Japan; [3]National Metrology Institute of Japan (NMIJ), National Institute of Advanced Industrial Science and Technology (AIST), Tsukuba, Japan; [4]Division of Applied Life Sciences, Graduate School of Agriculture, Kyoto University, Kyoto, Japan; [5]Department of Life Science and Applied Chemistry, Graduate School of Engineering, Nagoya Institute of Technology, Nagoya, Japan; [6]Department of Biological Sciences, Graduate School of Science, The University of Tokyo, Tokyo, Japan; [7]Applied Chemistry and Bioscience, Chitose Institute of Science and Technology, Chitose, Japan; [8]NODAI Genome Research Center, Tokyo University of Agriculture, Tokyo, Japan; [9]Laboratory of Aquatic Molecular Biology and Biotechnology, Graduate School of Agricultural and Life Sciences, University of Tokyo, Tokyo, Japan; [10]Department of Bioscience, Tokyo University of Agriculture, Tokyo, Japan; [11]Graduate School of Life and Environmental Sciences, University of Tsukuba, Tsukuba, Japan

*For correspondence:
ryo-futahashi@aist.go.jp

**Abstract** Many animals change their body color for visual signaling and environmental adaptation. Some dragonflies show wax-based color change and ultraviolet (UV) reflection, but the biochemical properties underlying the phenomena are totally unknown. Here we investigated the UV-reflective abdominal wax of dragonflies, thereby identifying very long-chain methyl ketones and aldehydes as unique and major wax components. Little wax was detected on young adults, but dense wax secretion was found mainly on the dorsal abdomen of mature males of *Orthetrum albistylum* and *O. melania*, and pruinose wax secretion was identified on the ventral abdomen of mature females of *O. albistylum* and *Sympetrum darwinianum*. Comparative transcriptomics demonstrated drastic upregulation of the *ELOVL17* gene, a member of the fatty acid elongase gene family, whose expression reflected the distribution of very long-chain methyl ketones. Synthetic 2-pentacosanone, the major component of dragonfly's wax, spontaneously formed light-scattering scale-like fine structures with strong UV reflection, suggesting its potential utility for biomimetics.
DOI: https://doi.org/10.7554/eLife.43045.001

## Introduction

Many organisms exhibit a variety of body color patterns for visual communication and environmental adaptation. The diversity of the color patterns encompasses the ultraviolet (UV) range, reflecting the fact that many animals can detect UV light as well as green, blue and/or red light (*Osorio and Vorobyev, 2008*). UV reflection has been reported from numerous organisms and may be important not only for protection against UV-induced damage but also for visual signaling (*Silberglied, 1979*;

**eLife digest** Humans have often looked to nature for answers to problems. Living things has evolved for millions of years to deal with life's challenges, and so engineers and inventors faced with similar challenges can also take inspiration from the natural world. Several plants and animals, for instance, reflect ultraviolet light. This ability may protect them from some of the damaging effects of sunlight; materials with similar properties would have a range of uses, including as coatings on windows that protect our homes and furniture or as cosmetics that protect ourselves in the same way.

Some dragonflies – including the white-tailed skimmer, which is particularly common in Japan – are partly coated with a wax that reflects both ultraviolet and visible light. These insects can also see ultraviolet light, which means it is likely that they also use the reflective wax to send visual signals to one another. However, the biochemistry of this wax and the genes involved in its production remained poorly understood.

Futahashi et al. have now found that the dragonfly wax consists mostly of very long-chain molecules known as methyl ketones and aldehydes; neither of which are a common components of other waxes. The wax was found in distinct patches on the bodies of adults; these patches were colored white with a hint of blue, while the rest of the dragonfly was mostly brown. Looking at gene activity in different parts of the dragonflies showed that a gene called *ELOVL17* is much more active in the wax-coated areas. This gene encodes an enzyme that makes long-chain molecules, and its activity closely matched the distribution of the especially long-chain methyl ketones on the dragonflies' surface.

Futahashi et al. then synthesized the major component of the surface wax – specifically, a chemical called 2-pentacosanone – in the laboratory, and saw that it spontaneously formed fine, scale-like structures that strongly reflected ultraviolet light. Further work is now needed to explore the potential applications of this bio-inspired wax, and to understand exactly what the dragonflies use it for in the wild.

DOI: https://doi.org/10.7554/eLife.43045.002

*Eaton and Lanyon, 2003*; *Paul and Gwynn-Jones, 2003*; *Lee, 2007*). Previous studies on biological UV reflection have focused on its optical properties and structural bases, such as multilayer surface structures (*Sun et al., 2013*). In some plants and insects, the production and secretion of wax on their surface has been reported to increase UV reflection (*Clark and Lister, 1975*; *Pope, 1979*; *Holmes and Keiller, 2002*; *Kakani et al., 2003*).

Dragonflies (including damselflies) are colorful, large-eyed, diurnal and actively flying insects, whose body colors often differ markedly between sexes, developmental stages, and closely related species (*Tillyard, 1917*; *Corbet, 1999*; *Futahashi et al., 2012*; *Futahashi, 2016*; *Futahashi, 2017*; *Bybee et al., 2016*). Because dragonflies are able to perceive UV light (*Bybee et al., 2012*; *Futahashi et al., 2015*), it seems plausible that UV color also plays important roles in their mate recognition, in male–male competition and in other ecological characteristics such as habitat selection and behavioral differences. Several studies have reported that the presence of a pruinose wax layer on the body surface accounts for UV reflection patterns in dragonflies (*Silberglied, 1979*; *Robertson, 1984*; *Hilton, 1986*; *Gorb, 1995*; *Harris et al., 2011*), but the biochemical properties and molecular composition of the wax, and the genes involved in wax production by dragonflies, are totally unknown.

Here, we mainly focus on the white-tailed skimmer dragonfly (*Orthetrum albistylum*), which is one of the most common dragonfly species in Japan (*Sugimura et al., 2001*; *Ozono et al., 2012*). As sexual maturity is reached, adult males of *O. albistylum* show wax-based body color change from light brown to blueish white, whereas adult females remain brownish throughout most of their lifetime, although very aged females become slightly whitish (*Figure 1A*) (*Sugimura et al., 2001*; *Ozono et al., 2012*). Notably, androchrome females, whose body color is very similar to that of adult males, have been recorded, though very rarely, in the field (*Figure 1A*) (*Sugimura et al., 2001*; *Ozono et al., 2012*). Androchrome females can be distinguished from very aged females because

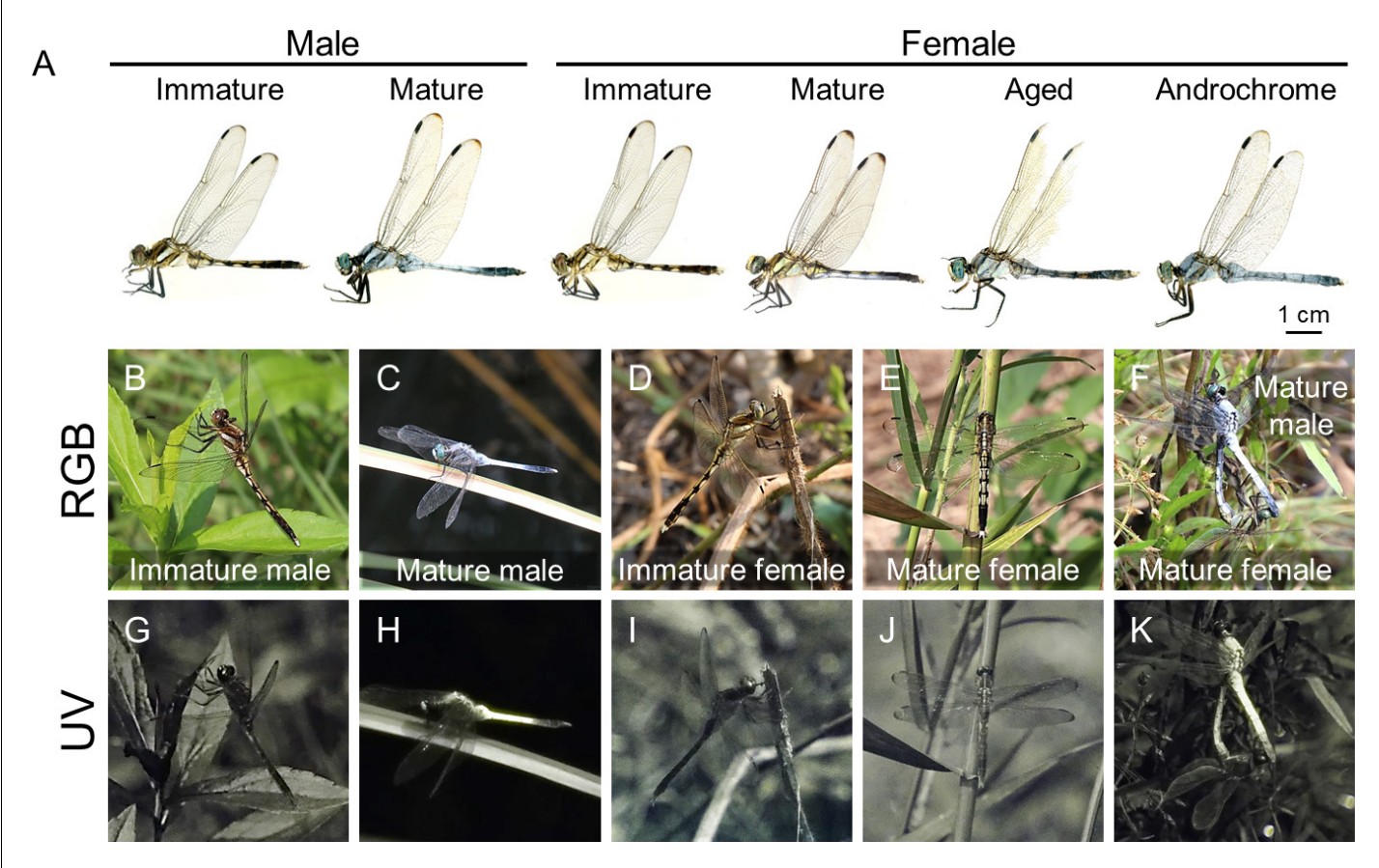

**Figure 1.** Stage- and sex-dependent adult color change in *O. albistylum* as visualized in red, green, blue (RGB) and ultraviolet (UV) light. (**A**) Adult males and females of *O. albistylum*. (**B**, **G**) Immature male. (**C**, **H**) Mature male. (**D**, **I**) Immature female. (**E**, **J**) Mature female. (**F**, **K**) Mating pair. Images photographed normally (**B–F**) or through a UV filter (**G–K**) in the field.
DOI: https://doi.org/10.7554/eLife.43045.003

the dorsal abdomen is more whitish than the ventral abdomen after the semimature stages (*Sugimura et al., 2001*; *Ozono et al., 2012*).

In this study, we investigated the ultrastructure, reflectance, wettability, chemical composition, self-organization, and biosynthesis pathway of surface wax in *O. albistylum* and allied dragonfly species. We found that, during the maturation process, adult males secrete a strongly light-scattering wax layer onto their body surface, thereby increasing their visibility not only in the blue and green wavelength ranges but also in the UV range. Chemically, the UV-reflective surface wax consisted of very long-chain methyl ketones and aldehydes, which have not been previously identified as major wax components. Comparative transcriptomics identified a gene encoding a member of the elongation of very long-chain fatty acids (ELOVL) protein family, whose expression was strongly correlated with the distribution of very long-chain methyl ketones on the surface of dragonflies. Notably, chemically synthesized 2-pentacosanone, the major component of the surface wax, spontaneously formed scale-like fine structures that strongly reflected UV light. These results provide a previously undescribed molecular and structural basis for wax-based body color change and UV reflection that has ecological and applied relevance.

## Results and discussion

### Stage- and sex-dependent body color change and UV reflection in *O. albistylum*

We compared the wax-based body color changes and UV reflection patterns of adult insects of *O. albistylum* using a high-sensitivity camera with a UV filter. UV reflection was hardly detected on the body surface of immature males and females (*Figure 1B,D,G and I*; *Video 1*). As sexual maturation proceeded, males accumulated whitish wax, mainly on their dorsal abdomen, which strongly reflected UV (*Figure 1C,F,H and K*; *Video 1*). It should be noted that in mature females, UV-reflective whitish wax was secreted on the ventral abdomen only (*Figure 1D,F,I and K*; *Video 1*). As adult aging proceeded further, not only males but also females developed pruinose wax on the entire body surface (*Figure 1A*), which resulted in considerable UV reflection even in females. Optical measurements of reflectance on both the dorsal and ventral abdominal regions were used for quantitative evaluation of sex- and stage-dependent changes in the adult insects of *O. albistylum*. Immature males and females mainly reflected light of above 500 nm in wavelength, and did not exhibit remarkable UV reflection (*Figure 2A and D*; *Figure 2—figure supplement 1A and D* ; *Figure 2— source data 1*; *Figure 2—source data 2*). In mature males, reflectance increased, in particular of light of wavelengths below 600 nm, resulting in strong UV reflection on the dorsal abdomen and moderate UV reflection on the ventral abdomen (*Figure 2B*; *Figure 2—figure supplement 1B* ; *Figure 2—source data 3*; *Figure 2—source data 4*). In mature females, by contrast, moderate UV reflection was observed on the ventral abdomen only (*Figure 2E*; *Figure 2—figure supplement 1E*; *Figure 2—source data 3*; *Figure 2—source data 4*). In aged males and females, reflectance increased to some extent on both the dorsal and the ventral sides of the abdomen (*Figure 2C and F*; *Figure 2—figure supplement 1C and F* ; *Figure 2—source data 5*; *Figure 2—source data 6*). Micro-spectrometry of small areas (10 µm x 10 µm) on the dorsal abdomen of a mature male indicated that the surface wax is responsible for overall reflectance, in particular in the short wavelength range including UV (*Figure 2G*; *Figure 2—source data 7*): strong reflection was found in the wax-covered white micro-areas (*Figure 2G; a, c, d, e and f*), whereas little reflection was detected in the blackish micro-areas where the surface wax was lost (*Figure 2G; b, g and h*).

### Surface fine structure of *O. albistylum*

Sex- and stage-dependent changes in the surface fine structure of *O. albistylum*, with special attention to the surface wax, were observed by scanning electron microscopy (SEM). In mature males, the dorsal abdomen was covered with scale- or plate-like fine structures (2–3 µm wide, 50 nm thick), which represented the secreted wax layer (*Figure 3A–D and K*). The depth of the wax layer reached up to 6 µm from the cuticle surface (*Figure 3D*). In mature females and immature individuals, by contrast, only tiny nanopillar-like structures (100 nm wide, 200–300 nm high) were seen on the dorsal abdomen (*Figure 3E–J and L*). In both mature males and females, on the other hand, small plate-like structures (up to 2 µm wide) were observed on the ventral abdomen, which presumably represented the pruinose wax secretions (*Figure 3O and P*) that were not conspicuous in immature males and females (*Figure 3M and N*). Here, we suggest that these fractal surface structures, consisting of randomly arranged fine wax platelets, are responsible for the whitish structural color that results from light scattering, as observed on the dorsal abdomen of mature males and the ventral abdomen of mature males and females. The idea that the secreted wax layer produces structural color was confirmed by a simple experiment: the whitish color disappeared when light scattering was disturbed by acetone application, and the whitish color instantly recovered upon evaporation of the applied acetone (*Video 2*). Note that the sizes of the wax platelets (up to 2–3 µm) are larger than the

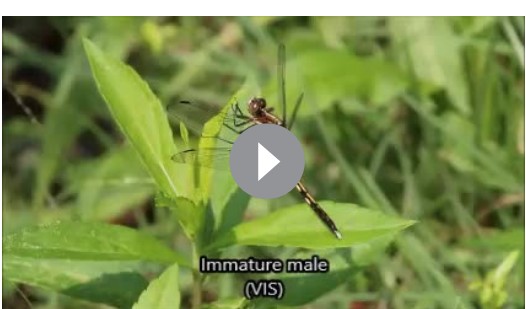

**Video 1.** Movie of visible light and UV reflection of *O. albistylum* in the field.
DOI: https://doi.org/10.7554/eLife.43045.013

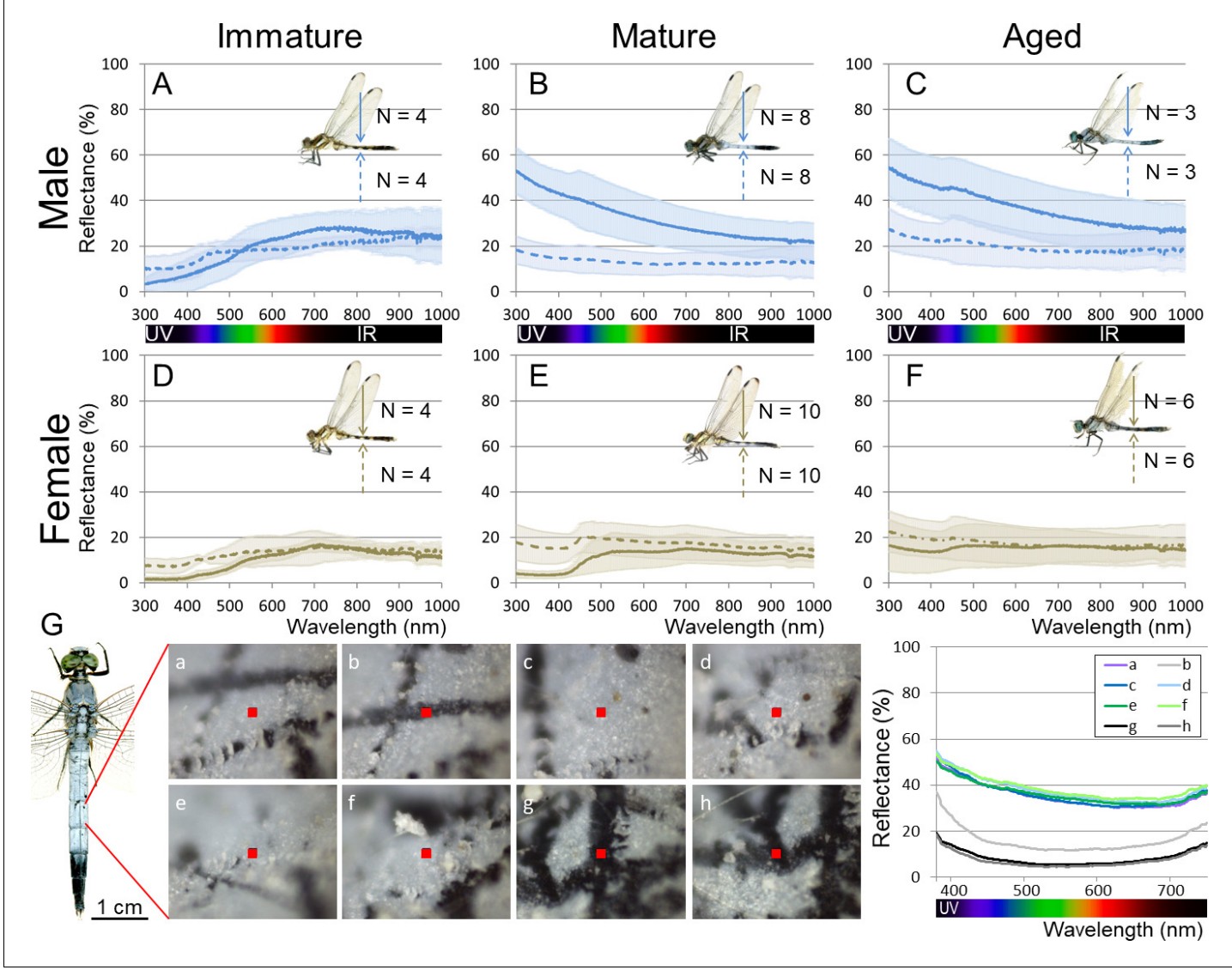

**Figure 2.** Reflectance of the adult body surface at the 5th abdominal segment of *O. albistylum*. (A–F) Spectrometry of a round area (6 mm in diameter) in males (A–C) and females (D–F). (A, D) Immature individuals. (B, E) Mature individuals. (C, F) Aged individuals. Solid and dotted lines indicate averaged UV reflectance on the dorsal and ventral sides of the abdomen, respectively. The standard deviation is shaded. (G) Micro-spectrometry of the 5th abdominal segment of a mature male. UV reflectance was measured in eight micro-areas (10 μm x 10 μm each) depicted as red squares in the photos. In the photos, white areas are covered with secreted wax whereas black areas are without wax, presumably because of accidental scratches and cracks on the adult body surface.

DOI: https://doi.org/10.7554/eLife.43045.004

The following source data and figure supplement are available for figure 2:

**Source data 1.** Spectrometry of a round area (6 mm in diameter) of immature males and females of *O. albistylum*.
DOI: https://doi.org/10.7554/eLife.43045.006

**Source data 2.** Spectrometry of a round area (6 mm in diameter) of mature males and females of *O. albistylum*.
DOI: https://doi.org/10.7554/eLife.43045.007

**Source data 3.** Spectrometry of a round area (6 mm in diameter) of aged males and females of *O. albistylum*.
DOI: https://doi.org/10.7554/eLife.43045.008

**Source data 4.** Micro-spectrometry of the 5th abdominal segment of a mature male of *O. albistylum*.
DOI: https://doi.org/10.7554/eLife.43045.009

**Source data 5.** Micro-spectrometry of immature males and females of *O. albistylum*.
DOI: https://doi.org/10.7554/eLife.43045.010

**Source data 6.** Micro-spectrometry of mature males and females of *O. albistylum*.

*Figure 2 continued*

DOI: https://doi.org/10.7554/eLife.43045.011

**Source data 7.** Micro-spectrometry of aged males and females of *O. albistylum*.

DOI: https://doi.org/10.7554/eLife.43045.012

**Figure supplement 1.** Micro-spectrometry (of 10 × 10 μm micro-areas) of *O. albistylum*.

DOI: https://doi.org/10.7554/eLife.43045.005

wavelength of UV and visible light and thus these platelets are capable of light scattering, while the sizes of the nanopillar structures (up to 200–300 nm) are smaller than the wavelength of UV light and thus incapable of light scattering (*Vukusic and Sambles, 2003*), and thus might cause a descrease in the reflectivity of the surface (*Kinoshita and Yoshioka, 2005*).

## Cuticular fine structure of *O. albistylum*

Specialized glands or structures for wax secretion have been characterized in diverse insects (*Pope, 1979*; *Ammar et al., 2015*), but such wax-producing structures have not been described from dragonflies. Transmission electron microscopic (TEM) observations of the abdominal ultrathin sections of *O. albistylum* identified a number of fine ducts penetrating the cuticle layer in both immature and mature individuals (*Figure 3Q–X*). On the dorsal abdomen of mature males in particular, the cuticular ducts were well-developed and full of electron-dense material, probably reflecting the active wax secretion there (*Figure 3S*, arrowheads).

## Chemical composition of dragonfly UV-reflective wax

To investigate the biochemical properties and molecular composition of dragonfly wax, the surface wax of *O. albistylum* was tested for solubility in organic solvents with reference to surface fine structure and wettability. We found that the secreted wax is insoluble in ethanol (*Figure 4B*), partially soluble in hexane (*Figure 4C and E*), and completely soluble in chloroform (*Figure 4D and F*). The untreated abdominal surface exhibited strong water repellency (*Figure 4A*), but removal of the wax by hexane or chloroform treatment resulted in drastically reduced water repellency or increased wettability (*Figure 4C and D*). As reported in a variety of plants and insects (*Hadley, 1981*), the strong water repellency conferred by the surface wax may be important for in reducing water loss from dragonflies that have an aerial lifestyle. On the basis of these results, we extracted the surface wax of *O. albistylum* with hexane or chloroform, and analyzed its chemical composition by gas chromatography and mass spectrometry. In the hexane extract from the dorsal abdomen of mature males, only three very long-chain methyl ketones, namely 2-pentacosanone ($C_{25}H_{50}O$), 2-heptacosanone ($C_{27}H_{54}O$) and 2-nonacosanone ($C_{29}H_{58}O$), were identified (*Figure 5A*; *Figure 5—figure supplement 1* ). In the chloroform extract from the dorsal abdomen of mature males, in addition to the three very long-chain methyl ketones, four very long-chain aldehydes, namely tetracosanal ($C_{24}H_{48}O$), hexacosanal ($C_{26}H_{52}O$), octacosanal ($C_{28}H_{56}O$), and triacosanal ($C_{30}H_{60}O$), were detected (*Figure 5B*; *Figure 5—figure supplement 1*). It should be noted that, in mature males, the very long-chain methyl ketones were dominant on the dorsal abdomen whereas the very long-chain aldehydes were dominant on the ventral abdomen (*Figure 5B and C*). By contrast, only the four very long-chain aldehydes were identified in the chloroform extract from the ventral abdomen of mature females (*Figure 5E*): neither the very long-chain methyl ketones nor the very long-chain aldehydes were found in the chloroform extract from the dorsal abdomen of mature females (*Figure 5D*). Such very long-chain methyl ketones and aldehydes have been identified, although not as major components, in the surface wax of plants (*Kolattukudy and Walton, 1972*; *Post-Beittenmiller, 1996*; *Yamamoto et al., 2008*; *Kunst and Samuels, 2003*) and in the skin lipids of snakes (*Ahern and Downing, 1974*). Similar ketones were characterized as sex pheromones of cockroaches (*Nishida et al., 1976*) and snakes (*Parker and Mason, 2014*). In the light of previous studies on the wax secretions of various insects, in which hydrocarbons, long-chain esters, alcohols, and/or free fatty acids were identified as major components (*Brown, 1975*; *Blomquist and Jackson, 1979*; *Hadley, 1981*), it seems that the chemical composition of the dragonfly's abdominal UV-reflective surface wax is unique among insects.

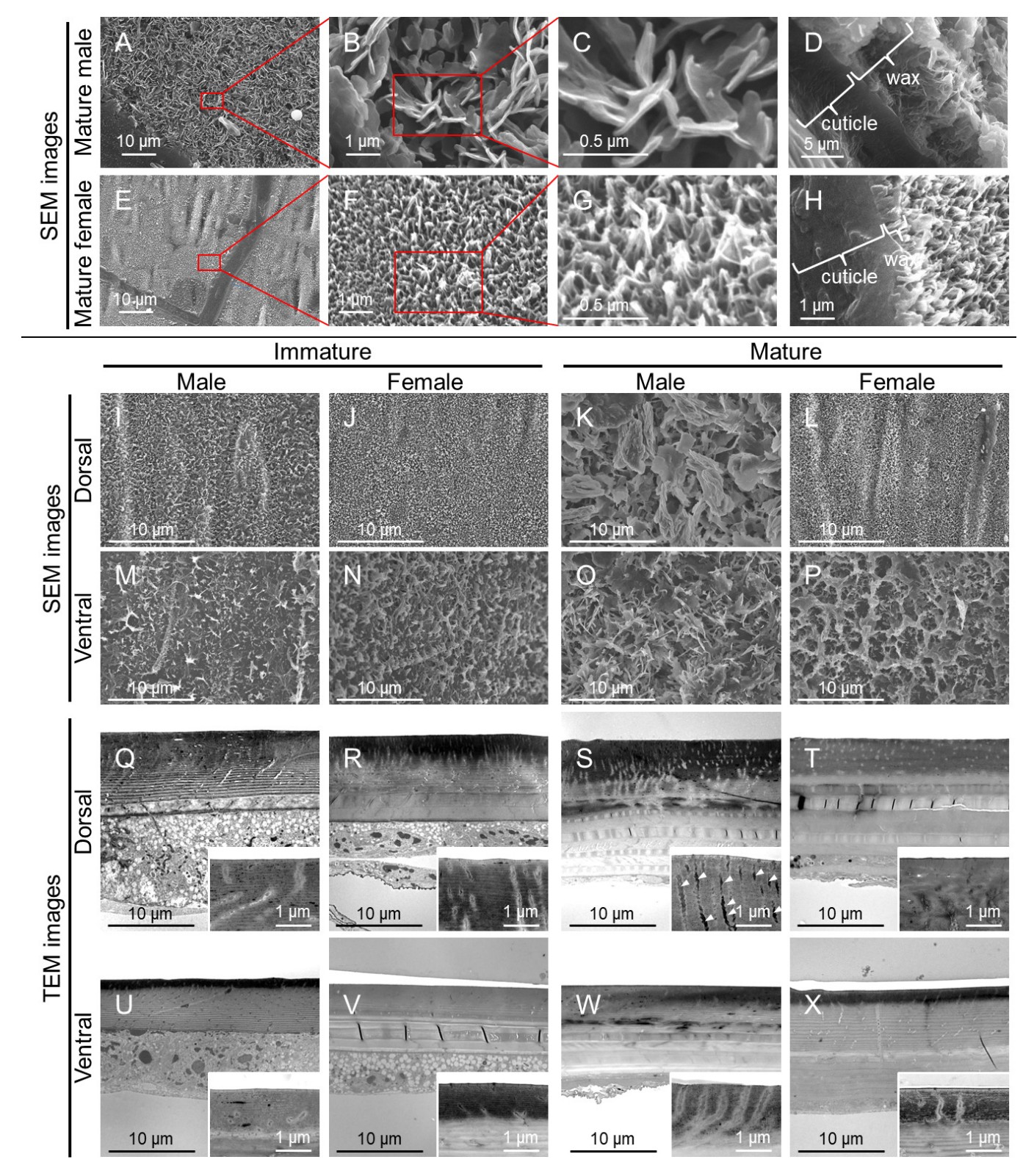

**Figure 3.** Fine structure of the adult body surface at the 5th abdominal segment of *O. albistylum*. (A–P) Scanning electron microscopic (SEM) images of the body surface. (Q–X) Transmission electron microscopic (TEM) images of the sectioned cuticle. It should be noted here that the surface wax was dissolved and removed during the processing of the sample for TEM observation. (A–H) Dorsal side of mature male (A–D) and mature female (E–H). Panels B and F are magnified images of panels A and E, as indicated by red rectangles. Likewise, panels C and G are magnified images of panels B

*Figure 3 continued on next page*

*Figure 3 continued*

and F. (D, H) Cross-sectioned images of cuticle and surface wax. (I–L and Q–T) Dorsal side. (M–P and U–X) Ventral side. (I, M, Q, U) Immature male. (J, N, R, V) Immature female. (K, O, S, W) Mature male. (L, P, T, X) Mature female.

DOI: https://doi.org/10.7554/eLife.43045.014

## Synthetic 'dragonfly wax' exhibits structural self-organization, strong reflection and water repellency

We chemically synthesized 2-pentacosanone, the main UV-reflective wax component identified from the dorsal abdomen of mature males of *O. albistylum* (*Figure 5F*), and attempted to recrystallize it on glass plates using three methods, namely dropping, quenching, and slow cooling (see Materials and methods). The dropping method yielded numerous wax platelets randomly arranged on the substratum (*Figure 6B and F*), which were reminiscent of the fine structure of the surface wax on the dorsal abdomen of mature males (*Figure 6A and E*). By contrast, the quenching method and the slow cooling method resulted in larger wax platelets (*Figure 6C,D,G and H*), which looked structurally dissimilar to dragonfly surface wax (*Figure 6A and E*). The 2-pentacosanone sheets made by the three methods showed qualitatively similar reflectance patterns across UV to visible range, which were also similar to the reflectance pattern shown by dragonfly surface wax (*Figure 6I–L*-source data 1). Notably, however, the 2-pentacosanone sheets made by the dropping method yielded stronger light reflectance and lower wettability than those made by the quenching method and the slow-cooling method (*Figure 6I–P*), probably because the smaller wax platelets made by the dropping method better mimic the fine structure of dragonfly surface wax. These results strongly suggest that the light-scattering nanostructure that is spontaneously formed by the secreted very long-chain methyl ketones, including 2-pentacosanone, should play a pivotal role in the formation of the wax layer that strongly reflects UV and visible light on the dorsal abdomen of mature males of *O. albistylum*.

## UV-reflective wax production in other dragonflies

Diverse dragonfly species are known to secrete whitish or bluish wax on their body surface (*Tillyard, 1917*; *Corbet, 1999*; *Sugimura et al., 2001*; *Ozono et al., 2012*). We investigated the wax production, reflectance, and chloroform-extracted wax composition of the abdominal body surface of three other dragonfly species, *Orthetrum melania*, *Sympetrum darwinianum*, and *Crocothemis servilia* (*Figure 7*). The blue-tailed skimmer dragonfly (*O. melania*), which is closely related to *O. albistylum*, prefers shady habitats in contrast to *O. albistylum* that tends to form territories in sunny places (*Sugimura et al., 2001*; *Ozono et al., 2012*). Mature males of *O. melania* develop bluish wax mainly on the dorsal abdomen, whereas mature females do not secrete wax even on the ventral abdomen (*Figure 7A*). In *O. melania*, UV reflection was observed only on the wax-bearing mature males (*Figure 7E,I and L*; *Figure 7—figure supplement 1A and D*; *Figure 7—source data 1*; *Figure 7—source data 2*; *Video 3*). In the red dragonfly *S. darwinianum*, only mature females secrete whitish wax on the ventral abdomen (*Figure 7B*), where UV reflection was clearly detected (*Figure 7F,J and M*; *Figure 7—figure supplement 1B and E*; *Figure 7—source data 3*; *Figure 7—source data 4*; *Video 3*). The scarlet dragonfly *C. servilia* develops little wax on its body surface throughout its lifetime (*Figure 7C and D*), and no UV reflection was detected in either sex (*Figure 7G,H,K and N*; *Figure 7—figure supplement 1C and F*; *Figure 7—source data 5*; *Figure 7—source data 6*). These reflectance data indicated that (i) the presence of the secreted wax on the body surface accounts for the UV-reflecting body regions in all dragonfly species, (ii) the levels of the UV reflection vary among dragonfly species and also across different body regions, and (iii) the dorsal abdomen of mature males of *O. albistylum* exhibits the

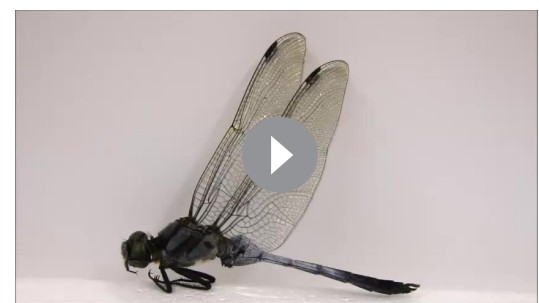

**Video 2.** Reversible color change by adding acetone.
DOI: https://doi.org/10.7554/eLife.43045.015

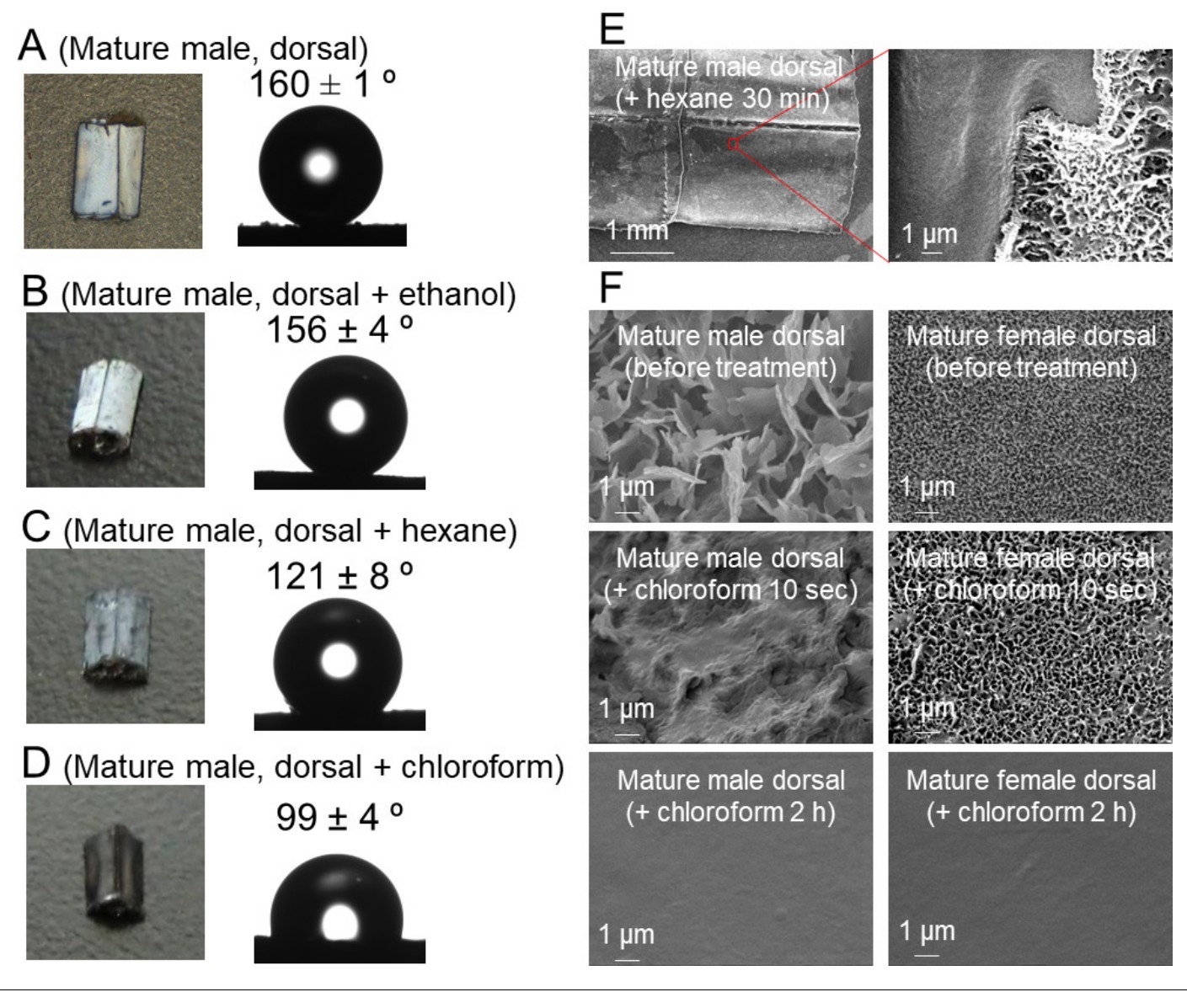

**Figure 4.** Solubility and wettability of the abdominal wax of *O. albistylum*. (A–D) Dorsal side of the 5th abdominal segment of mature males. (A) No treatment. (B) After ethanol treatment. (C) After hexane treatment. (D) After chloroform treatment. (E) Scanning electron microscope images of the dorsal surface of mature male 30 min after hexane treatment. (F) Scanning electron microscope images of dorsal side of a mature male (left) or a mature female (right) after chloroform treatment.

DOI: https://doi.org/10.7554/eLife.43045.016

strongest UV reflection (*Figures 2* and *7*). The wax composition also varied among dragonfly species (*Figure 8*). For example, only very long-chain aldehydes were detected from mature males of *O. melania* (*Figure 8A and C*), whereas only a very long-chain aldehyde, tetracosanal, was identified from the ventral abdomen of mature females of *S. darwinianum* (*Figure 8B and C*). Very long-chain methyl ketones were detected only in regions where UV reflection was strong in *O. albistylum* (*Figure 8C*).

## Ecological relevance of UV-reflective wax production in dragonflies

Here we point out that the UV reflection and wax production patterns observed in these dragonflies seem to reflect, at least to some extent, their environmental and behavioral characteristics. In general, body pigmentation and/or wax production are conspicuous in mature dragonflies, especially

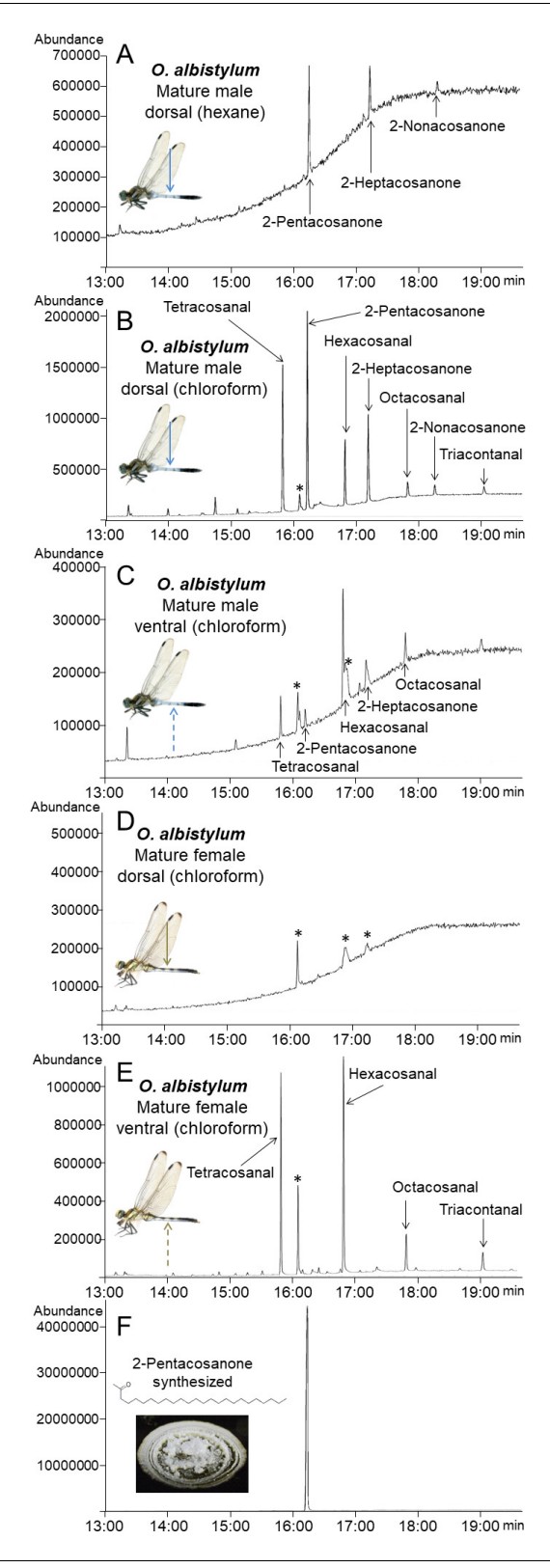

**Figure 5.** Identification and chemical synthesis of surface wax of *O. albistylum*. (A) Chromatogram of hexane extract from the dorsal abdomen of a mature male. (B) Chromatogram of chloroform extract from the dorsal abdomen of a mature male. (C) Chromatogram of chloroform extract from the ventral abdomen of a mature male. (D) Chromatogram of chloroform extract from the dorsal abdomen of a mature female. (E) Chromatogram of

*Figure 5 continued on next page*

*Figure 5 continued*

chloroform extract from the ventral abdomen of a mature female. (**F**) Chromatogram of chemically synthesized 2-pentacosanone. Asterisks indicate nonspecific peaks that are also detected with the solvent only. The Y-axis shows the abundance of total ion current.

DOI: https://doi.org/10.7554/eLife.43045.017

The following figure supplement is available for figure 5:

**Figure supplement 1.** Mass spectra and extracted ion chromatogram of dragonfly wax

DOI: https://doi.org/10.7554/eLife.43045.018

in reproductively active territorial males (*Tillyard, 1917*), which may be related to aspects of mate recognition, male–male competition, UV protection and anti-desiccation protection that are fatally important for them (*Corbet, 1999*). In the closely related *Orthetrum* species, *O. albistylum* dominates sunny habitats and shows stronger UV reflection than *O. melania,* which prefers shady habitats: mature males of *O. albistylum* form a dense whitish wax layer on their dorsal abdomen (*Figure 1C and F*; *Figure 9A*) whereas mature males of *O. melania* develop a relatively thin bluish wax layer (*Figure 7A*; *Figure 9B*); mature females of *O. albistylum* wear pruinose wax on their ventral abdomen (*Figure 1F*) whereas mature females of *O. melania* do not (*Figure 7A*). Similar patterns are found in other *Orthetrum* species: mature males of *O. luzonicum* prefer sunny habitats and develop whitish wax (*Figure 9C*), whereas mature males of *O. glaucum* are associated with shady habitats and form bluish wax (*Figure 9D*). It is intriguing to ask why the mature females of some dragonflies, such as in *O. albistylum* (*Figure 1F and K*) and *S. darwinianum* (*Figure 7B and F*), produce UV-reflective wax only on their ventral abdomen. We point out that, in these species, males form territories in sunny places, wait for females that fly in, and chase and copulate with them on nearby plants (*Figure 1F*; *Figure 7B*). Therefore, these species usually mate in sunny places, where the female's ventral abdomen is exposed to direct sunshine for an extended period (*Figure 1F*; *Figure 7B*). By contrast, *O. melania* usually mates in shady places (*Figure 7A*) and *C. servilia* quickly mates during flight for only a few seconds (*Figure 7D*), and the female's ventral abdomen develops no wax and exhibits no UV reflection in these species (*Figure 7A,D,E and H*). On the basis of these observations, we speculatively suggest that the female's ventral wax might protect the ventral abdomen, which is less pigmented, less sclerotized and containing ovaries, against UV damage.

## Comparative transcriptomics of wax-producing and non-wax-producing epidermal regions in *O. albistylum*

What are the molecular mechanisms that underlie the production of dragonfly surface wax of unique chemical composition? Notably, rarely discovered gynandromorphic dragonflies consistently exhibit discontinuous surface wax patterns (*Figure 9E and F*) (*Sugimura et al., 2001*; *Karube and Machida, 2015*), suggesting that de novo wax production may be regulated in a cell-autonomous manner. In an attempt to gain insights into the molecular basis of dragonfly wax production, we performed RNA sequencing using samples of the dorsal and ventral abdominal epidermis dissected from immature, semimature, mature, mature-aged, and aged individuals of both sexes of *O. albistylum*. The adult maturity was judged on the basis of the amount of wax and the wing condition. In addition, we were able to examine a mature androchrome female (*Figure 10A*). *Figure 10B* summarizes the RNA sequencing data. A total of 7790 genes whose maximum fragments per kilobase of transcript per million mapped reads (FPKM) values were greater than 10, 1708 exhibited doubled expression levels or significantly higher expression in the dorsal epidermis of mature males (with wax) when compared with that in the dorsal epidermis of mature females (no wax); 518 genes exhibited doubled expression levels or significantly higher expression in the dorsal epidermis of an androchrome female (with wax) when compared with that in the dorsal epidermis of mature females (no wax); and 305 genes were commonly identified in these two categories as upregulated genes associated with wax production. Of these 305 genes, 26 genes were highly expressed (FPKM >100), of which five genes exhibited extremely high expression levels (FPKM >1,000) (*Figure 10C*; *Figure 10—figure supplement 1*; accession nos. LC416763- LC416767).

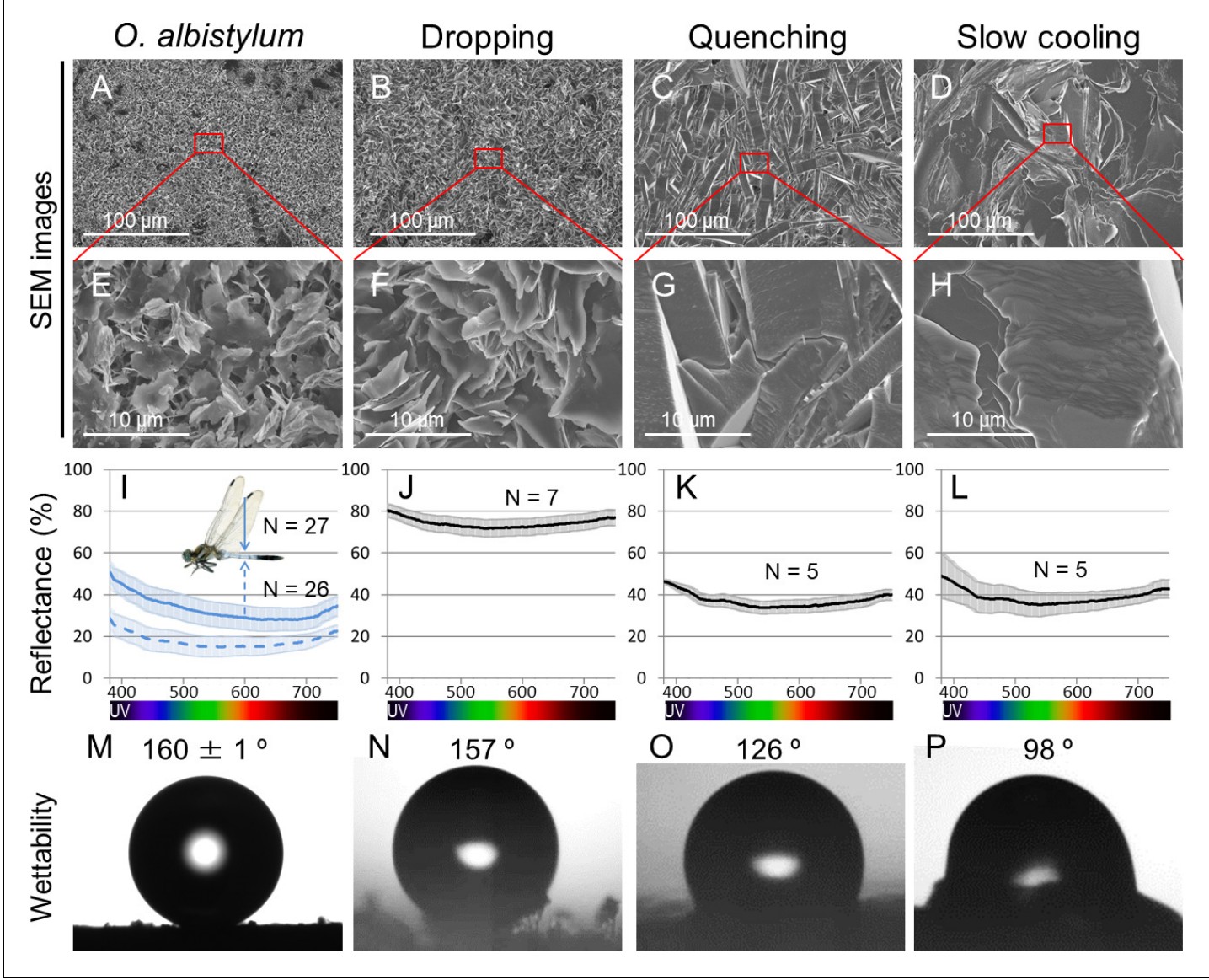

**Figure 6.** Comparison of surface fine structure, reflectance, and wettability between the dorsal abdomen of mature males of *O. albistylum* and synthetic 2- pentacosanone crystallized on glass plates using three different cooling processes. (A–H) Scanning electron microscope images. Panels E, F, G and H are magnified images of panels A, B, C and D, respectively, as indicated by red rectangles. (I–L) Micro-spectrometry from a 10 µm x 10 µm area. (M–P) Wettability measured with a 1 nL water droplet. (A, E, I, M) The dorsal abdomen of mature males of *O. albistylum*. (B, F, J, N) Synthetic wax crystallized by the dropping method. (C, G, K, O) Synthetic wax crystallized by the quenching method. (D, H, L, P) Synthetic wax crystallized by the slow-cooling method.

DOI: https://doi.org/10.7554/eLife.43045.019

The following source data is available for figure 6:

**Source data 1.** Micro-spectrometry of synthetic 2-pentacosanone crystallized on glass plates using three different cooling processes.
DOI: https://doi.org/10.7554/eLife.43045.020

## Drastically upregulated ELOVL genes in wax-producing epidermis of *O. albistylum*

Among the extremely upregulated genes, we identified a gene belonging to the elongation of very long-chain fatty acids (ELOVL) protein family that was drastically and specifically (more than 100-fold) expressed in the dorsal epidermis of semimature and mature males and the androchrome female (*Figure 10C and D*; *Figure 10—figure supplement 1* ). ELOVL proteins catalyze the

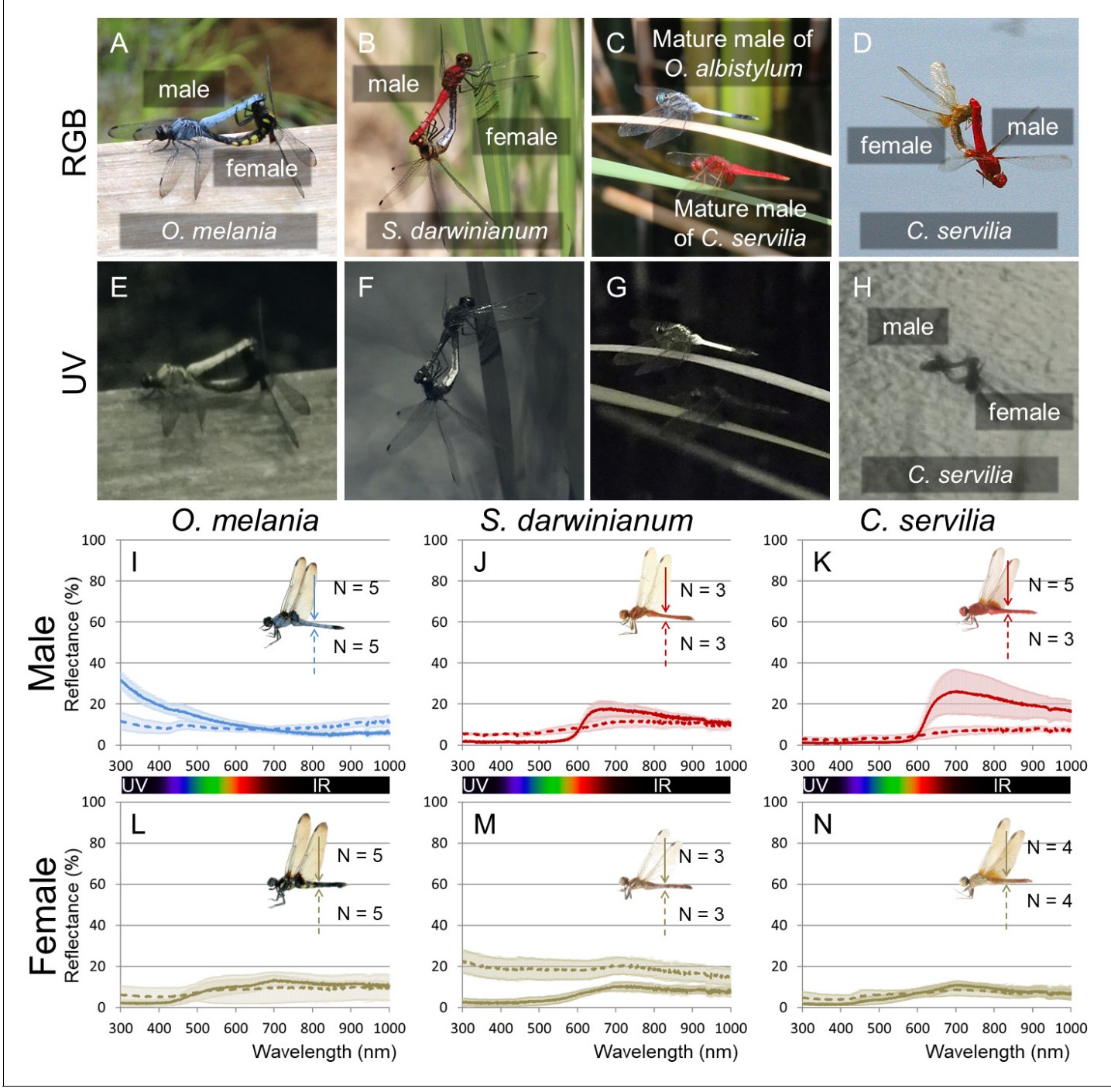

**Figure 7.** UV reflection patterns in *O. melania*, *S. darwinianum* and *C. servilia*. (**A, E**) Mating pair of *O. melania*. (**B, F**) Mating pair of *S. darwinianum*. (**C, G**) Mature male of *C. servilia* with mature male of *O. albistylum*. (**D, H**) Mating pair of *C. servilia*. Each image photographed normally (RGB) (**A–D**) or through a UV filter (**E–H**) in the field. (**I–N**) Spectrometry of a round area (6 mm in diameter) on the 5[th] abdominal segment of *O. melania* (**I, L**), *S. darwinianum* (**J, M**), and *C. servilia* (**K, N**). (**I–K**) Male. (**L–N**) Female. Solid and dotted lines indicate averaged UV reflectance on the dorsal and ventral sides of the abdomen, respectively. The standard deviation is shaded.

DOI: https://doi.org/10.7554/eLife.43045.021

The following source data and figure supplement are available for figure 7:

**Source data 1.** Spectrometry of a round area (6 mm in diameter) of *O. melania*.
DOI: https://doi.org/10.7554/eLife.43045.023

**Source data 2.** Spectrometry of a round area (6 mm in diameter) of *S. darwinianum*.

*Figure 7 continued on next page*

*Figure 7 continued*

DOI: https://doi.org/10.7554/eLife.43045.024

**Source data 3.** Spectrometry of a round area (6 mm in diameter) of *C. servilia*.

DOI: https://doi.org/10.7554/eLife.43045.025

**Source data 4.** Micro-spectrometry of *O. melania*.

DOI: https://doi.org/10.7554/eLife.43045.026

**Source data 5.** Micro-spectrometry of *S. darwinianum*.

DOI: https://doi.org/10.7554/eLife.43045.027

**Source data 6.** Micro-spectrometry of *C. servilia*.

DOI: https://doi.org/10.7554/eLife.43045.028

**Figure supplement 1.** Micro-spectrometry (of a 10 × 10 µm micro-area) of *O. melania* (A, D), *S. darwinianum* (B, E), and *C. servilia* (C, F).

DOI: https://doi.org/10.7554/eLife.43045.022

elongation of fatty acids with acyl chains longer than 18 carbon atoms (**Kihara, 2012**) and also of hydrocarbons (**Chertemps et al., 2007**). ELOVL proteins are conserved from yeast to mammals, and 3, 7 or 20 ELOVL family protein genes are identified in the genomes of yeast, mammals or the fruit fly *Drosophila melanogaster*, respectively (**Figure 11A**) (**Szafer-Glusman et al., 2008**; **Kihara, 2012**). In the transcriptomic data of *O. albistylum* and also in the draft genome data of the scarce chaser dragonfly (*Ladona fulva*) (BCMHGSC: I5K, GenBank accession no. APVN00000000.2), we identified 17 ELOVL genes in total (**Figure 11A**; accession nos. BR001497-BR001513, LC416747-LC416763), all of which contained a conserved HXXHH motif (**Figure 11B**).

In addition to the drastically upregulated ELOVL gene (=*ELOVL17*) mentioned above, two ELOVL genes exhibited notable upregulation patterns (**Figure 11C**). *ELOVL14* was highly expressed in the dorsal epidermis of semimature and mature males, and also in both the dorsal and the ventral epidermis of the androchrome female (**Figure 10E**; **Figure 11**). In aged females, *ELOVL17* and *ELOVL14* genes were slightly upregulated (**Figure 10D and E**), which may account for the slight wax secretion of aged females of *O. albistylum* (see **Figure 1A**). Meanwhile, the *ELOVL3* gene was preferentially expressed in the ventral abdomen of immature females (**Figure 10F**), which may be relevant to the preferential accumulation of very long-chain aldehydes on the ventral abdomen of mature females of *O. albistylum* (see **Figure 5E**). These results strongly suggest that these ELOVL genes are involved in production of the surface wax of dragonflies, which mainly consists of very long-chain methyl ketones and aldehydes. To confirm this idea, we attempted to knock-down the expression level of the *ELOVL17* gene by injection of small interfering RNA followed by electroporation, a technique established in other dragonfly species (**Okude et al., 2017**). After employing the RNAi treatment, we expected that the wax production on the abdominal surface of mature males would be suppressed. Unfortunately the electroporation damaged the adult cuticle and caused high mortality of the treated insects, so we failed to observe the phenotypes expected for the RNAi experiment. Hence, the precise functions of the dragonfly's ELOVL gene products remain to be verified in future studies, for which the establishment of a stable laboratory rearing system for *O. albistylum* and the successful application of genome-editing technology in this species are anticipated. In

addition to the *ELOVL17* gene, the *Acyl-CoA Delta(11) desaturase*, *ferritin*, *NPC intercellular transporter 2*, and *uncharacterized protein* (LC416764- LC416767) genes exhibited extremely high expression in wax-producing regions, although their stage and region specificity was not prominent compared to those of the *ELOVL17* gene. The biological roles of these genes also deserve future studies.

## Conclusion and perspective

In this study, we found that mature males of *O. albistylum* exhibit strong light reflection, including the reflection of light in the UV range, because of a previously uncharacterized

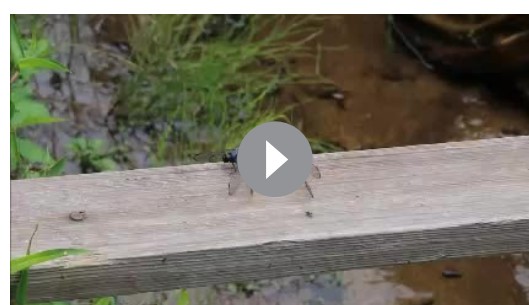

**Video 3.** Movie of visible light and UV reflection of *O. melania*, *S. darwinianum*, and *C. servilia* in the field.

DOI: https://doi.org/10.7554/eLife.43045.030

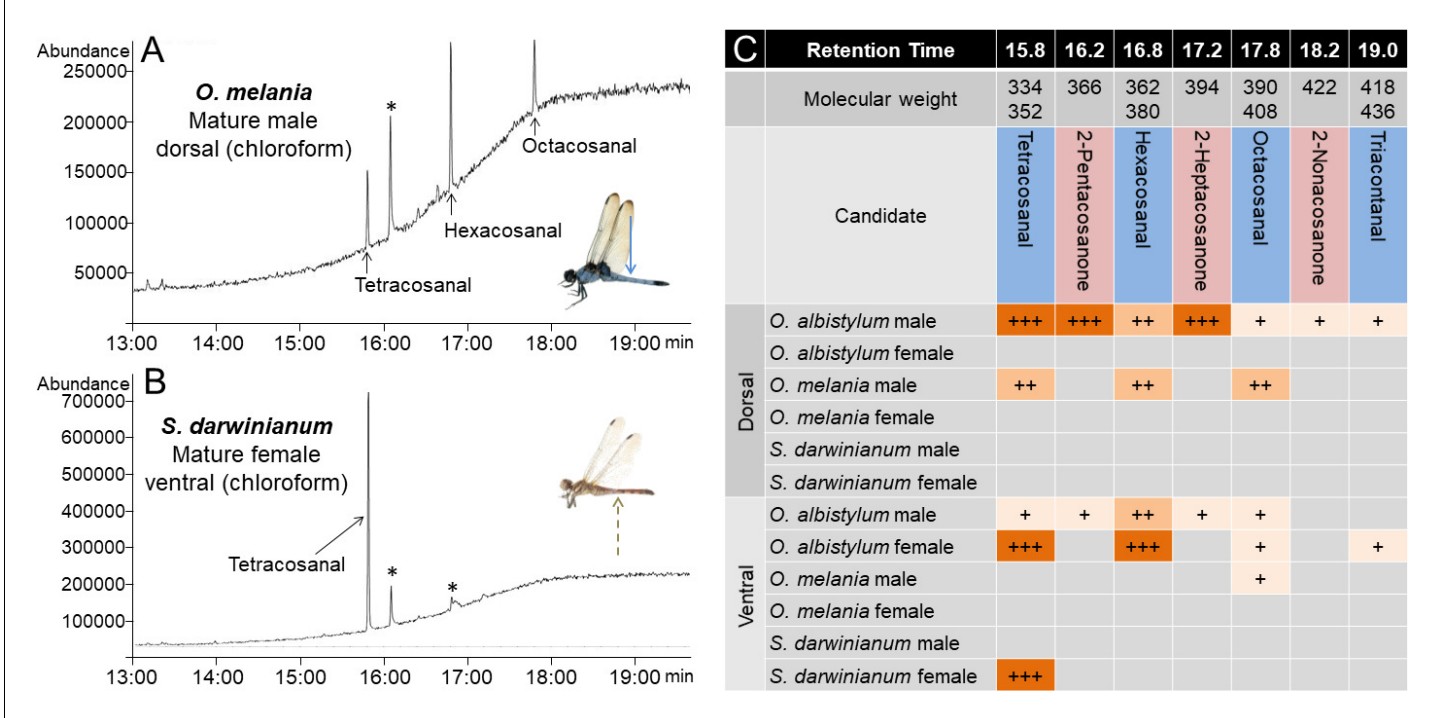

**Figure 8.** Comparison of wax components on the dorsal and ventral abdomen of *O. albistylum*, *O. melania* and *S. darwinianum*. (A) Chromatogram of chloroform extract from the dorsal abdomen of a mature male of *O. melania*. (B) Chromatogram of chloroform extract from the ventral abdomen of a mature female of *S. darwinianum*. Asterisks indicate nonspecific peaks also detected with the solvent only. (C) Summary of wax components detected from *O. albistylum*, *O. melania* and *S. darwinianum*. Relative amount was judged from the abundance of total ion current. +++, high amount; ++, moderate amount; +, small amount.

DOI: https://doi.org/10.7554/eLife.43045.029

mechanism, namely very long-chain methyl ketone production. Plausibly, differences in wax production between sexes, stages and species are important for signal communications between dragonflies, and may reflect their habitats and behavior. It should be noted that synthesized 2-pentacosanone, a major component of very long-chain methyl ketone, reproduced the strong reflection, surface fine structure, and water repellency. Considering that UV reflective materials can be applied in the fields of cosmetics and painting, and that *O. albistylum* has been traditionally used as medicine in Asia (*Corbet, 1999*), the dragonflies' UV reflective wax may have the potential to be utilized as a novel eco-friendly biological material.

## Materials and methods

### Insects and UV images

Adult insects of *O. albistylum*, *O. melania*, *S. darwinianum,* and *C. servilia* were collected at Tsukuba, Ibaraki, Japan, or at Imizu, Toyama, Japan. Photos of UV reflection were taken in the field using a high-sensitivity camera (Sony A7S, IDAS UV-VIS mod) and a UV filter (IDAS-U).

### Spectrometry and micro-spectrometry

In order to investigate the wax-based color change quantitatively, the dorsal and ventral parts of abdominal segment 5 were surgically divided and used for reflection measurements. The reflections from small areas (diameter 6 mm) were taken using a spectrometer (HR2000+, Ocean Optics), and those of micro areas (10 × 10 μm) were carried out using a micro-spectrometer (CRAIC Technologies) equipped with an upright microscope (Eclipse E-400; Nikon). The specimens were epi-illuminated with a 75 W Xenon arc lamp (Nikon), and the measurements were obtained from an approximately 10 μm x 10 μm area. The reflected spectral radiances were converted to relative

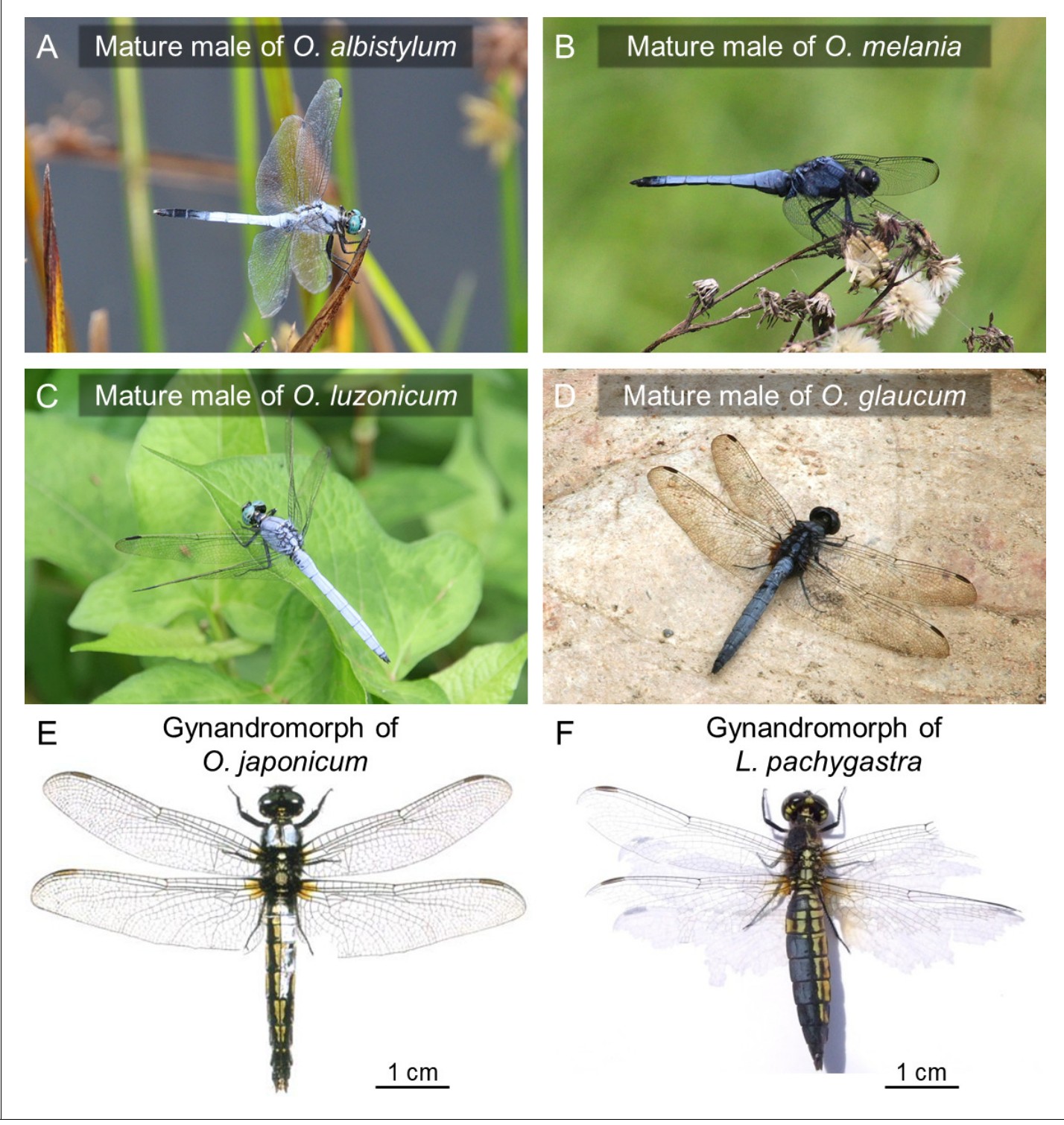

**Figure 9.** Surface wax of *Orthetrum* species and allied dragonflies. (A–D) Mature males of *O. albistylum* (A), *O. melania* (B), *O. luzonicum* (C) and *O. glaucum* (D). (E–F) Gynandromorphic individuals of *O. japonimum* (E) and *Lyriothemis pachygastra* (F). Photos courtesy of Mitsutoshi Sugimura (E) and Makoto Machida (F).

DOI: https://doi.org/10.7554/eLife.43045.031

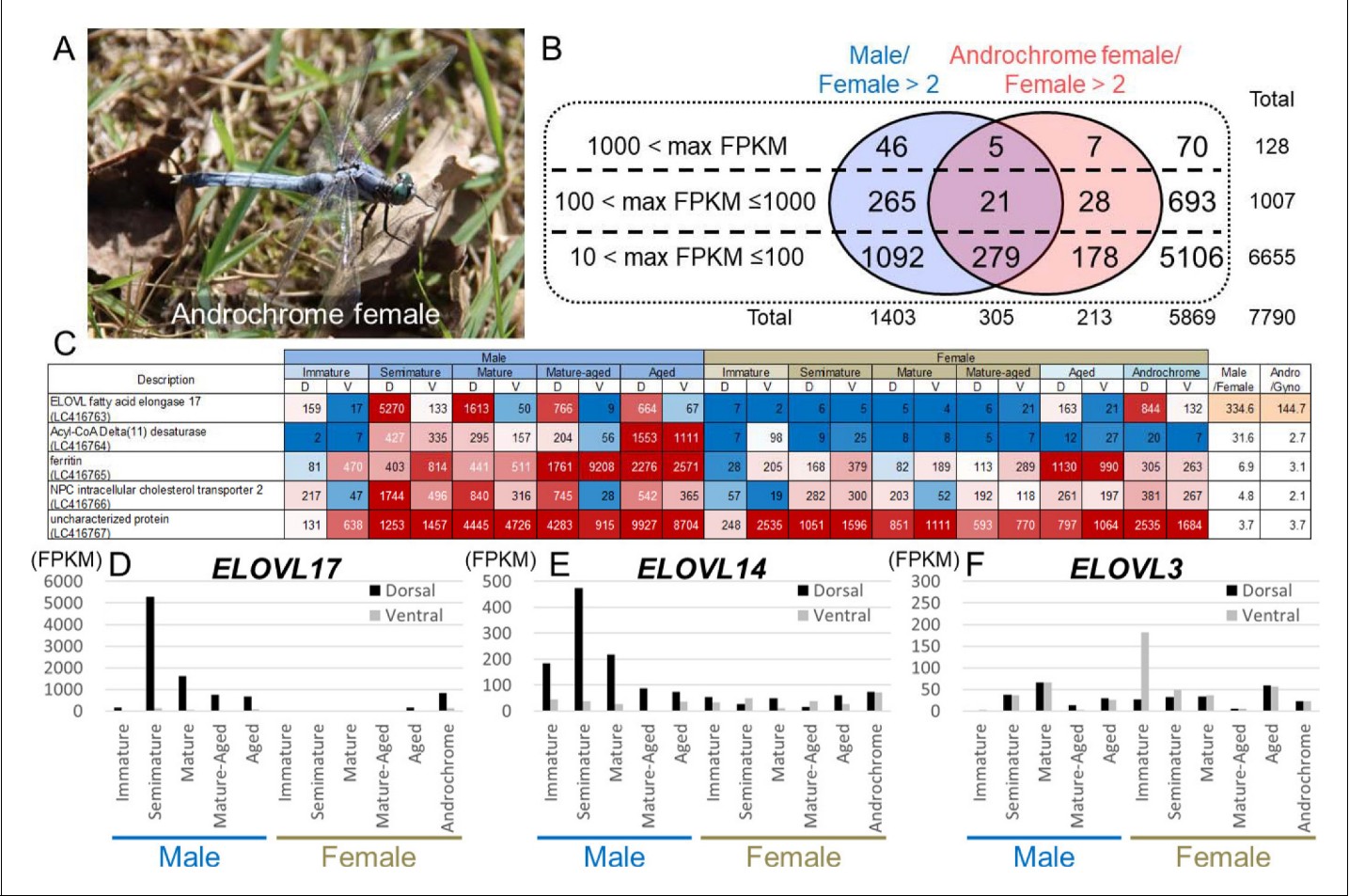

**Figure 10.** Genes associated with UV reflective wax. (**A**) An androchrome female used for transcriptome analysis. (**B**) The number of genes that were upregulated in the dorsal abdominal epidermis of males and/or an androchrome female compared to normal females. (**C**) The list of genes upregulated in the dorsal abdominal epidermis of both males and an androchrome female (max FPKM >1000). Gene expression levels are displayed as a heat map. The numbers indicate FPKM values. Red and blue fills indicate high and low expression levels, respectively. D and V indicate the dorsal and ventral abdominal regions, respectively. (**D–F**) Expression level of three elongation of very long-chain fatty acids (ELOVL) genes in the dorsal and ventral parts of the epidermis of *O. albistylum*.

DOI: https://doi.org/10.7554/eLife.43045.032

The following figure supplement is available for figure 10:

**Figure supplement 1.** List of genes that are upregulated in the dorsal abdominal epidermis of both males and an androchrome female (max FPKM >100).

DOI: https://doi.org/10.7554/eLife.43045.033

reflectance by normalization with a white reflectance standard (Spectralon USRS-99–010, Labsphere).

## Histology

The microstructural changes resulting from wax secretion were examined by scanning electron microscopy (SEM) and transmission electron microscopy (TEM). For SEM observation, the dissected dorsal and ventral parts of abdominal segment 5 were coated with a 2–3 nm osmium layer using hollow-cathode plasma chemical vapor deposition (HPC-1SW; Vacuum Device). They were then observed under a scanning electron microscope (H-4800; Hitachi) with an accelerating voltage of 5 kV. For TEM observation, dissected dorsal and ventral parts of abdominal segment 5 were prefixed for 12 hr in 2% glutaraldehyde and 2% paraformaldehyde in 0.1 M cacodylate buffer (pH7.2), post-fixed with 1% osmium tetroxide for 2 hr in 0.1 M cacodylate buffer, and embedded in Quetol 812-

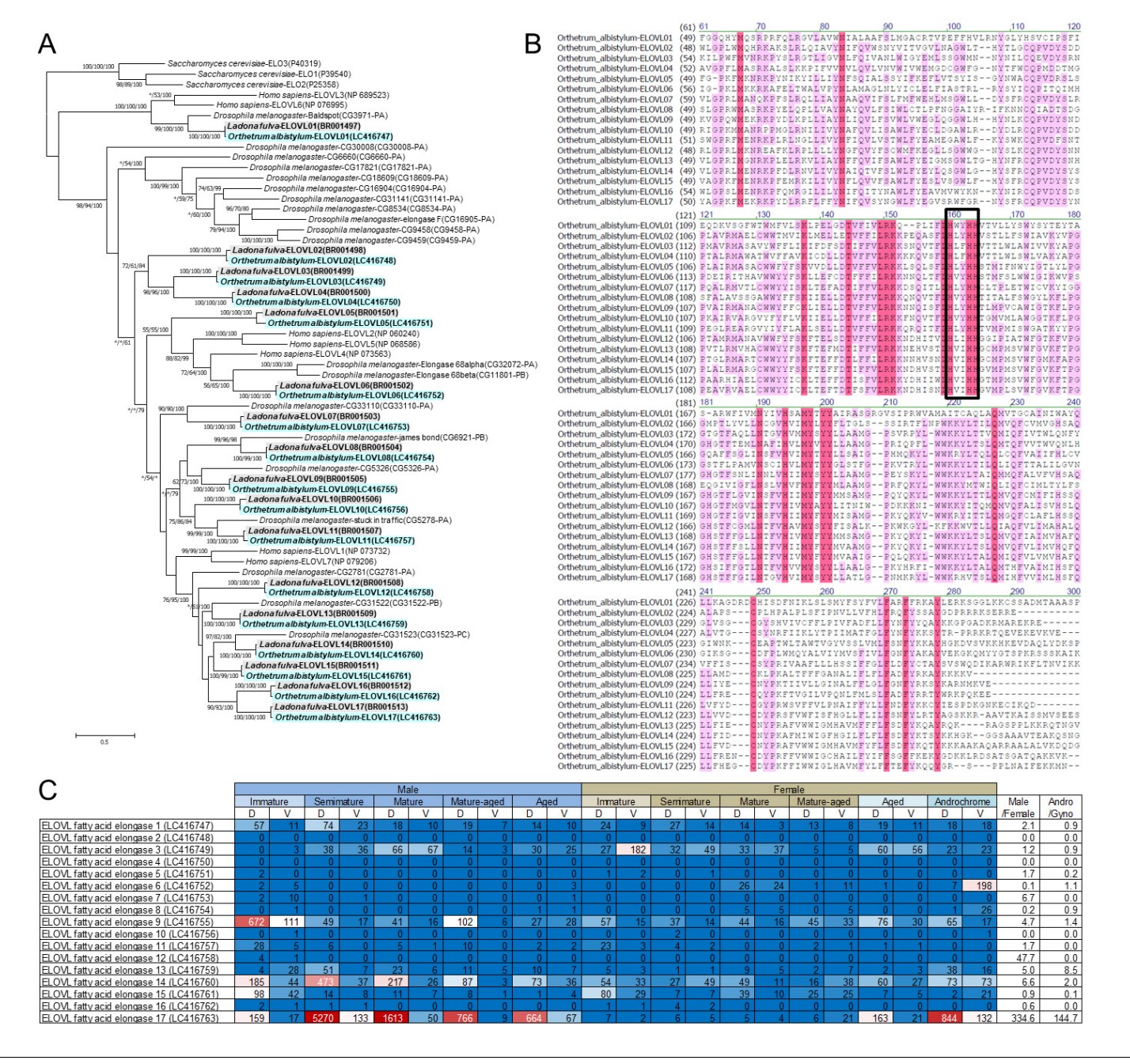

**Figure 11.** Identification of 17 ELOVL genes in dragonflies. (**A**) Phylogenetic tree of ELOVL family genes produced on the basis of their amino-acid sequences. A maximum likelihood phylogeny is shown, but neighbor-joining and Bayesian phylogenies exhibit substantially the same topologies. Statistical supporting values are indicated for each node (shown as (bootstrap value of neighbor-joining)/(bootstrap value of maximum likelihood)/ (posterior probability of Bayesian)). Asterisks indicate support values < 50%. Blue and gray shading indicates *O. albistylum* and *L. fulva* genes, respectively. Accession numbers or annotation identities are shown in parentheses. (**B**) Alignment of 17 ELOVL genes of *O. albistylum*. The conserved histidine motif is boxed. (**C**) Gene expression levels of 17 ELOVL genes in *O. albistylum*. The numbers indicate FPKM values. Red and blue shading indicates high and low expression levels, respectively. D and V indicate the dorsal and ventral abdominal regions, respectively.
DOI: https://doi.org/10.7554/eLife.43045.034

Araldite regin mixture (Nisshin EM). Ultrathin sections (approximately 70 nm thick) were cut perpendicular to the anterior–posterior axis on an ultra-microtome (UC7; Leica) with a diamond knife (DiA-TOME), stained with 2% uranyl acetate for 5 min followed by lead citrate solution for 3 min (Sigma-Aldrich), and observed under a transmission electron microscope (JEM-1400; JEOL, 100 KV).

## Measurement of surface wettability and wax solubility

To elucidate the biochemical properties of dragonfly wax, surface wettability and solubility were examined. Surface wettability was evaluated on the basis of the contact angle of water micro-droplet on the samples. Each sample was fixed on a glass substrate, and a micro-droplet of distilled water (about 1.0 nL) was placed on the surface of the sample. The shape of droplet was recorded immediately using a high-speed camera (HAS-220; Ditect) with a microscopic contact angle meter (MCA-3; Kyowa Interface Science). Wax solubility was analyzed by treating the dissected abdominal segment 5 with hexane or chloroform for 30 min.

## Wax extraction and analysis

To identify the molecular composition of dragonfly wax, wax samples were extracted from living specimens with chloroform or hexane. To avoid contamination from internal lipids during gas chromatography/mass spectrometry (GC-MS) analysis, the solvent was carefully pipetted several times onto the abdominal surface of the living individuals. The extracts were analyzed by GC-MS using a 6890N GC coupled with 5973 MSD (Agilent) in the split-less mode, using a DB-5MS fused silica column (30 m x 0.25 mm i.d., 0.25 µm film thickness, Agilent) with helium as the carrier gas at a flow rate of 1.0 mL/min at a temperature programmed to change from 80°C (1 min) to 320°C at a rate of 15 °C/min and then held for 3 min. The mass spectrometer was operated in the scan mode with 70 eV ionazation voltage as electron ionization. Histological examinations, wettability tests, and GC-MS analyses were conducted using different samples.

## Chemical synthesis of wax components

To confirm whether the very long-chain methyl ketones form the scale-like fine structures, 2-pentacosanone, the major component of dragonfly wax was chemically synthesized from 1-tetracosanol via 1-tetracosanal and 2-pentacosanol. Pyridinium chlorochromate (PCC, 1.29 g, 5.97 mmol) was added to a suspension of 1-tetracosanol (395 mg, 1.11 mmol) and powdered molecular sieves 4A (2.5 g) in dry $CH_2Cl_2$ (35 mL), and stirred for 4 hr at room temperature. The mixture was filtered through Celite and washed with diethyl ether. The combined filtrate and washings were filtered through florisil (15 g), washed with diethyl ether (200 mL) and concentrated in vacuo. The residue was chromatographed on silica gel (15 g) and concentrated in vacuo to give a white solid of 1-tetracosanal (290 mg, 0.82 mmol, 74%, GC $t_R$ = 23.7 min, MS $m/z$ (%) = 352 ($M^+$, 2), 334 (18), 96 (78), 82 (100), 57 (93), 43(72)). A solution of 1-tetracosanal (176 mg, 0.50 mmol) in dry tetrahydrofuran (THF, 10 mL) was cooled in ice bath. When the temperature reached 0°C, 1.4 M $CH_3MgBr$ in THF:toluene 1:3 (1 mL, 1.4 mmol) was added dropwise, and stirred for 1.5 hr at 0°C and for 1 hr at room temperature. The reaction was quenched with saturated $NH_4Cl$ (5 mL), and the product was extracted with hexane (3 × 20 mL). The organic layer was dried with anhydrous magnesium sulfate and concentrated in vacuo. The residue was chromatographed on silica gel (15 g, ethyl acetate/hexane, 1:5) to give a white solid of 2-pentacosanol (106 mg, 0.29 mmol, 58%). PCC (335 mg, 1.56 mmol) was added to a suspension of 2-pentacosanol (257 mg, 0.70 mmol) and powdered molecular sieves 4A (1.0 g) in dry $CH_2Cl_2$ (20 mL), and stirred for 3 hr at room temperature. The mixture was filtered through Celite and washed with diethyl ether. The combined filtrate and washings were filtered through florisil (15 g), washed with diethyl ether (200 mL) and concentrated in vacuo. The residue was recrystallized from hexane to give a white solid of 2-pentacosanone (202 mg, 0.55 mmol, 76%). Nuclear magnetic resonance (NMR) spectra of 2-pentacosanal and 2-pentacosanone were measured with Bruker AV-400 III Spectrometer (400 MHz) using tetramethylsilane (TMS) as an internal standard. Each sample was dissolved in $CDCl_3$ and $^1H$ spectrum was acquired. $^1H$-NMR of 2-pentacosanal ($CDCl_3$, 400 MHz) was as follows: δ 3.79 (1H, sex, $J$ = 6.0 Hz, H-2), δ 1.18 (3H, d, $J$ = 6.0 Hz, H-1), δ 0.88 (3H, t, $J$ = 6.0 Hz, H-25). $^1H$-NMR of 2-pentacosanone ($CDCl_3$, 400 MHz) is as follows: δ 2.41 (2H, d, $J$ = 7.6 Hz, H-3), δ 2.13 (3H, s, H-1), δ 0.88 (3H, t, $J$ = 6.4 Hz, H-25).

## Preparation of biomimetic wax surfaces

Biomimetic wax surfaces were composed of micro crystals of 2-pentacosanone. Heated 2-pentacosanone was recrystallized on gold-coated glass plates by different cooling processes: 1) continuous dropping of micro fused material of 1.0 µL under room temperature, 2) quenching from the melting state, and 3) maintenance near melting point at 64°C from the melting state. Micro-spectrometry, surface fine structure, and wettability were analyzed using a micro-spectrometer (CRAIC Technologies), a scanning electron microscope (H-4800; Hitachi), or a high-speed camera (HAS-220; Ditect) with a microscopic contact angle meter (MCA-3; Kyowa Interface Science), respectively, as described above.

## Transcriptome analysis

To investigate the genes involved in wax production, total RNA samples were extracted from the freshly dissected abdomens of *O. albistylum* using an RNeasy mini kit (Qiagen) or a Maxwell 16 LEV Simply RNA Tissue kit (Promega). RNA sequencing was performed as described previously (*Futahashi et al., 2015*). Using 1 µg of total RNA per sample as template, cDNA libraries were constructed using TruSeq RNA Sample Preparation Kits v2 (Illumina) and sequenced by HiSeq2000, Hiseq2500, or MiSeq (Illumina). The sequence data were deposited in the DNA Data Bank Japan Sequence Read Archive (accession numbers are shown in *Supplementary file 1*). The raw reads were subjected to de novo assembly using the Trinity program (*Grabherr et al., 2011*) implemented in the MASER pipeline (*Kinjo et al., 2018*). After automatic assembling, we checked and manually corrected the sequences of genes that are highly expressed in mature males using the Integrative Genomics Viewer (*Thorvaldsdóttir et al., 2013*) as reported previously (*Futahashi, 2017*). After revising the sequences, sequence read mapping was performed using the BWA-MEM program (*Li, 2013*) implemented in the MASER pipeline, whereby transcript expression levels were estimated in terms of fragments per kilobase per million reads (FPKM) values. ELOVL genes of *L. fulva* were obtained by a tBLASTn search against the draft genome sequence (APVN00000000.2) (https://www.hgsc.bcm.edu/).

## Phylogenetic analysis

To construct the molecular phylogeny of ELOVL family genes, deduced amino-acid sequences were aligned using the Clustal W program implemented in MEGA7 (*Kumar et al., 2016*). Molecular phylogenetic analyses were conducted by the neighbor-joining method and the maximum-likelihood method using MEGA7, and by the Bayesian method using MrBayes version 3.1.2 (*Ronquist et al., 2012*). Bootstrap values for neighbor-joining and maximum likelihood phylogenies were obtained by 1000 resampling. In total, 3750 trees were generated for each Bayesian analysis (ngen = 500,000, samplefreq = 100, burn in = 1250).

## Acknowledgements

We thank Tateo Shimozawa for SEM observation, Hiroyuki Futahashi for insect samples, Mitsutoshi Sugimura and Makoto Machida for the photos of gynandromorphic dragonflies in *Figure 9*, and Mutsuo Tanaka and Minoru Moriyama for technical assistance. We would like to acknowledge i5K, Stephen Richards and Oliver Niehuis for allowing us to use the *L. fulva* genome data. This work was supported by the Japan Society for the Promotion of Science Grant-in-Aid for Scientific Research Grants JP26660276, JP18H02491, and JP18H04893 (to RF), and by a Cooperative Research Grant of the Genome Research for BioResource, NODAI Genome Research Center, Tokyo University of Agriculture (to RF, RMK, and SY). This research was partially supported by the Platform Project for Supporting Drug Discovery and Life Science Research (Basis for Supporting Innovative Drug Discovery and Life Science Research (BINDS)) from AMED (Grant Number JP17am0101001).

## Additional information

### Competing interests

Ryo Futahashi, Yumi Yamahama, Migaku Kawaguchi, Daisuke Ishii, Ryouka Kawahara-Miki, Shunsuke Yajima, Takahiko Hariyama: An international patent on the synthesis method and application of very long chain methyl ketones and aldehydes was applied as PCT/JP2018/019559. The other authors declare that no competing interests exist.

### Funding

| Funder | Grant reference number | Author |
| --- | --- | --- |
| Japan Society for the Promotion of Science | JP26660276 | Ryo Futahashi |
| Genome research for BioResource NODAI Genome Research Center | | Ryo Futahashi Ryouka Kawahara-Miki Shunsuke Yajima |
| Japan Society for the Promotion of Science | JP18H02491 | Ryo Futahashi |
| Japan Society for the Promotion of Science | JP18H04893 | Ryo Futahashi |

The funders had no role in study design, data collection and interpretation, or the decision to submit the work for publication.

### Author contributions

Ryo Futahashi, Conceptualization, Resources, Data curation, Formal analysis, Supervision, Funding acquisition, Validation, Investigation, Visualization, Methodology, Writing—original draft, Project administration; Yumi Yamahama, Data curation, Formal analysis, Investigation, Methodology, Writing—original draft; Migaku Kawaguchi, Formal analysis, Investigation, Writing—original draft; Naoki Mori, Investigation, Methodology, Writing—original draft; Daisuke Ishii, Formal analysis, Investigation, Methodology, Writing—original draft; Genta Okude, Yuji Hirai, Data curation, Formal analysis, Investigation.; Ryouka Kawahara-Miki, Kazutoshi Yoshitake, Shunsuke Yajima, Data curation, Investigation; Takahiko Hariyama, Data curation, Investigation, Writing—original draft, Project administration; Takema Fukatsu, Supervision, Writing—original draft, Project administration

### Author ORCIDs

Ryo Futahashi (iD) http://orcid.org/0000-0003-4791-7054

### Decision letter and Author response

Decision letter https://doi.org/10.7554/eLife.43045.040
Author response https://doi.org/10.7554/eLife.43045.041

## Additional files

### Supplementary files

• Supplementary file 1. *Orthetrum albistylum* samples and RNA sequencing reads.
DOI: https://doi.org/10.7554/eLife.43045.035

• Transparent reporting form
DOI: https://doi.org/10.7554/eLife.43045.036

### Data availability

The sequences reported in this paper have been deposited in the DNA Data Bank Japan Read Archive, www.ddbj.nig.ac.jp (accession nos. BR001497-BR001513, LC416747-LC416767, DRA001687, DRA001690, DRA001693-DRA001694, DRA001697-DRA001698, DRA001700-DRA001701, DRA001703-DRA001704, DRA001706-DRA001707, DRA001709-DRA001710,

DRA001712-DRA001713, DRA001716-DRA001717, DRA007015-DRA007018). All data generated or analysed during this study are included in the manuscript and supporting files. Source data files have been provided for Figures 2,6I-L,7I-N, Figure 2-figure supplement 1, and Figure 7-figure supplement 1.

The following previously published dataset was used:

| Author(s) | Year | Dataset title | Dataset URL | Database and Identifier |
|---|---|---|---|---|
| Murali S, Richards S, Bandaranaike D, Bellair M, Blankenburg K, Chao H, Dinh H, Doddapaneni H, Dugan-Rocha S, Elkadiri S, Gnanaolivu R, Hernandez B, Skinner E, Javaid M, Lee S, Li M, Ming W, Munidasa M, Muniz J, Nguyen L, Hughes D, Osuji N, Pu L-L, Puazo M, Qu C, Quiroz J, Raj R, Weissenberger G, Xin Y, Zou X, Han Y | 2017 | Ladona fulva genome sequencing | https://www.ncbi.nlm.nih.gov/nuccore/APVN00000000.2 | NCBI Nucleotide, APVN00000000.2 |

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
