## [Decision Letter]

Thank you for submitting your article "Molecular basis of UV-reflective surface wax in dragonflies" for consideration by *eLife*. Your article has been reviewed by two peer reviewers, and the evaluation has been overseen by a Reviewing Editor and Patricia Wittkopp as the Senior Editor. The following individual involved in review of your submission has agreed to reveal his identity: Beatriz Willink (Reviewer #2).

The reviewers have discussed the reviews with one another and the Reviewing Editor has drafted this decision to help you prepare a revised submission.

Summary:

This is an integrative and complete study examining the structure, biochemical composition and proximate molecular mechanisms for the production of UV-reflective wax in dragonflies. In this well-written manuscript, the authors inquire into substantial gaps in our understanding of the structural coloration of insects and provide insights of relevance across fields, from animal physiology, evolution and ecology to material science. The study focuses on a common dragonfly species which produces sexually-dimorphic wax recovering different parts of the abdominal cuticle. The authors examine the reflective properties of this substance, its microscopic structure, chemical composition and other properties, to then identify key genes associated with its production. These results are discussed with reference to the ecological conditions that might have favored the evolution of such wax production, and its diversity across a few related species is also briefly touched upon. As such the manuscript should be of interest to entomologist, ecologist as well as physicist and material scientists working on biomimetic and bio-inspired (soft) matter properties.

Essential revisions:

1) Please broaden the scope of the light spectrum discussion. Whereas we appreciate that UV reflectance is the focus of your manuscript the wax may increase the general visibility, so beyond the UV spectrum too. An alternative interpretation of the photographs and the reflectance spectra is that the function of the wax is to increase the visibility more generally, by enhanced scattering, in a wide wavelength range, from the far-UV up to the infrared, i.e. throughout the full visible wavelength range of the dragonflies. It would be great if this perspective can be considered and discussed in the Introduction and Discussion. Whereas we agree that a focus on UV light makes sense from the perspective that not all animals use the UV spectrum, and it is thus of interest when the UV spectrum is used, the discussion becomes stronger when the vision of dragonflies is considered beyond the UV spectrum initially. This also provides context to clarify any limitations in the interpretation of the data. E.g. the present writing about UV-reflecting wax may suggest to some readers that there is a major difference of a factor 10 or so, but when we compare the abdomen reflectances at the peak wavelengths of the shortest and longest wavelength receptors, taking to be 340 and 520 nm, we estimate a 40% difference. So, framing the differences carefully with help the reader interprets the findings more accurately. Further, the value of the paper remains unequivocal when the message is broadened to include that males crucially increase their general visibility, in the UV as well as in the blue and green wavelength ranges, by secreting a strongly scattering wax layer. To better reflect this general relevance, we also recommend making the title more inclusive in case you agree. Otherwise, we would like you to carefully consider these revisions in the Discussion and conclusion as well as in the Abstract as a baseline.

2) The authors could do a better job at motivating their interesting study in the Introduction. Here, potential roles of UV reflectance for physical protection and communication are hinted, but for most readers it might not be immediately clear what can we gain from understanding how UV-reflectance is attained in this particular group of insects. Perhaps the authors could elaborate on what they refer to as "other ecological aspects" for which UV reflectance might important, whether in dragonflies or other animals. They could also address how understanding the chemical composition and structure of this wax can be useful to understand and replicate its functions, or more generally, to understand the diversity of mechanisms by which similar ecological functions can be attained in different organisms.

3) We think some sections in the Materials and methods would benefit from adding an initial sentence explaining the aim of the described analyses. For example, the objective of the phylogenetic analyses described in the Materials and methods was not clear to me until I had a look at the supplementary figures. We also could not find the legends for the supplementary figures, so several of their details were not entirely clear. Similarly, it should be clarified what is the aim of the wettability assay and how results in Figure 4A-D are interpreted.

---

## [Author Response]

Essential revisions:1) Please broaden the scope of the light spectrum discussion. Whereas we appreciate that UV reflectance is the focus of your manuscript the wax may increase the general visibility, so beyond the UV spectrum too. An alternative interpretation of the photographs and the reflectance spectra is that the function of the wax is to increase the visibility more generally, by enhanced scattering, in a wide wavelength range, from the far-UV up to the infrared, i.e. throughout the full visible wavelength range of the dragonflies. It would be great if this perspective can be considered and discussed in the Introduction and Discussion. Whereas we agree that a focus on UV light makes sense from the perspective that not all animals use the UV spectrum, and it is thus of interest when the UV spectrum is used, the discussion becomes stronger when the vision of dragonflies is considered beyond the UV spectrum initially. This also provides context to clarify any limitations in the interpretation of the data. E.g. the present writing about UV-reflecting wax may suggest to some readers that there is a major difference of a factor 10 or so, but when we compare the abdomen reflectances at the peak wavelengths of the shortest and longest wavelength receptors, taking to be 340 and 520 nm, we estimate a 40% difference. So, framing the differences carefully with help the reader interprets the findings more accurately. Further, the value of the paper remains unequivocal when the message is broadened to include that males crucially increase their general visibility, in the UV as well as in the blue and green wavelength ranges, by secreting a strongly scattering wax layer. To better reflect this general relevance, we also recommend making the title more inclusive in case you agree. Otherwise, we would like you to carefully consider these revisions in the Discussion and conclusion as well as in the Abstract as a baseline.

Thank you for the helpful comments. We changed the title to

"Molecular basis of wax-based color change and UV reflection in dragonflies"

We revised Abstract, Introduction, Results and Discussion as follows.

Abstract: "Many animals change their body color for visual signaling and environmental adaptation. Some dragonflies show wax-based color change and ultraviolet (UV) reflection, but biochemical properties underlying the phenomena are totally unknown. Here we investigated the UV-reflective abdominal wax of dragonflies…"

Introduction: "Many organisms exhibit a variety of body color patterns for visual communication and environmental adaptation. The diversity of the color patterns encompasses the ultraviolet (UV) range, reflecting the fact that many animals can detect UV light as well as green, blue and/or red (Osorio and Vorobyev, 2008). UV reflection has been reported from numerous organisms, which may be important not only for protection against UV-induced damage but also for visual signaling…"

"We found that, during the maturation process, adult males secrete a strongly light-scattering wax layer on the body surface, thereby increasing their visibility not only in the blue and green wavelength ranges but also in the UV range."

Results and Discussion: "We compared wax-based body color change and UV reflection patterns of adult insects of *O. albistylum*…"

"Immature males and females mainly reflected light above 500 nm in wavelength, and did not exhibit remarkable UV reflection…"

"In mature males, reflectance increased, in particular below 600 nm, which resulted in strong UV reflection on the dorsal abdomen and moderate UV reflection on the ventral abdomen…"

"Micro-spectrometry of small areas (10 µm x 10 µm) on the dorsal abdomen of a mature male indicated that the surface wax is responsible for overall reflectance, in particular in the short wavelength range including UV…"

Conclusion and Perspective: "In this study, we found that mature males of *O. albistylum* exhibit strong reflection including UV range in a previously uncharacterized way, namely very long-chain methyl ketone production."

Because peak wavelengths of the shortest and longest wavelength receptors remain unknown in *O. albistylum*, we did not mention about specific wavelengths.

2) The authors could do a better job at motivating their interesting study in the Introduction. Here, potential roles of UV reflectance for physical protection and communication are hinted, but for most readers it might not be immediately clear what can we gain from understanding how UV-reflectance is attained in this particular group of insects. Perhaps the authors could elaborate on what they refer to as "other ecological aspects" for which UV reflectance might important, whether in dragonflies or other animals. They could also address how understanding the chemical composition and structure of this wax can be useful to understand and replicate its functions, or more generally, to understand the diversity of mechanisms by which similar ecological functions can be attained in different organisms.

We revised these sentences as follows:

"Because dragonflies are able to perceive UV light (Bybee et al., 2012; Futahashi et al., 2015), it seems plausible that UV color also plays important roles in their mate recognition, male-male competition as well as other ecological conditions such as habitat selection and behavioral differences. Several studies have reported that a pruinose wax layer on the body surface accounts for UV reflection patterns in dragonflies (Silberglied, 1979; Robertson, 1984; Hilton, 1986; Gorb, 1995; Harris et al., 2011), but its biochemical properties, molecular composition, and genes involved in the dragonfly’s wax production have been totally unknown."

3) We think some sections in the Materials and methods would benefit from adding an initial sentence explaining the aim of the described analyses. For example, the objective of the phylogenetic analyses described in the Materials and methods was not clear to me until I had a look at the supplementary figures.

We added the initial sentence explaining the aim of the described analyses in the Materials and methods as follows:

"In order to quantitatively investigate the wax-based color change, the dorsal and ventral parts of abdominal segment 5 were surgically divided and used for reflection measurements."

"The microstructural changes due to wax secretion was examined by scanning electron microscopy (SEM) and transmission electron microscopy (TEM)."

"To elucidate the biochemical properties of dragonfly’s wax, surface wettability and solubility was examined."

"To identify the molecular composition of dragonfly’s wax, wax samples were extracted from living specimens with chloroform or hexane."

"To confirm whether the very long-chain methyl ketones form the scale-like fine structures, 2-pentacosanone, the major component of dragonfly’s wax was chemically synthesized from 1-tetracosanol via 1-tetracosanal and 2-pentacosanol."

"To investigate the genes involved in wax production, total RNA samples were extracted from the freshly dissected abdomens of *O. albistylum…*"

"To construct the molecular phylogeny of ELOVL family genes, deduced amino acid sequences were aligned…"

We also could not find the legends for the supplementary figures, so several of their details were not entirely clear.

We are sorry for forgetting the legends of the supplementary figures at the previous submission. We have now included these legends.

Similarly, it should be clarified what is the aim of the wettability assay and how results in Figure 4A-D are interpreted.

We added the following sentence.

"To investigate the biochemical properties and molecular composition of dragonfly’s wax, the surface wax of *O. albistylum* was tested for solubility in organic solvents with reference to surface fine structure and wettability."